# Coordinated integrin activation by actin-dependent force during T-cell migration

Pontus Nordenfelt[1,2,3], Hunter L. Elliott[2] & Timothy A. Springer[1]

For a cell to move forward it must convert chemical energy into mechanical propulsion. Force produced by actin polymerization can generate traction across the plasma membrane by transmission through integrins to their ligands. However, the role this force plays in integrin activation is unknown. Here we show that integrin activity and cytoskeletal dynamics are reciprocally linked, where actin-dependent force itself appears to regulate integrin activity. We generated fluorescent tension-sensing constructs of integrin $\alpha L\beta 2$ (LFA-1) to visualize intramolecular tension during cell migration. Using quantitative imaging of migrating T cells, we correlate tension in the $\alpha L$ or $\beta 2$ subunit with cell and actin dynamics. We find that actin engagement produces tension within the $\beta 2$ subunit to induce and stabilize an active integrin conformational state and that this requires intact talin and kindlin motifs. This supports a general mechanism where localized actin polymerization can coordinate activation of the complex machinery required for cell migration.

[1] Department of Biological Chemistry and Molecular Pharmacology, Harvard Medical School and Program in Cellular and Molecular Medicine, Children's Hospital Boston, 3 Blackfan Circle, Boston, Massachusetts 02115, USA. [2] Image and Data Analysis Core, Harvard Medical School, 240 Longwood Ave., Boston, Massachusetts 02115, USA. [3] Department of Clinical Sciences, Division of Infection Medicine, Faculty of Medicine, Lund University, BMC, B14, Sölvegatan 19, 22362 Lund, Sweden. Correspondence and requests for materials should be addressed to P.N. (email: nordenfelt@crystal.harvard.edu or pontus.nordenfelt@med.lu.se) or to T.A.S. (email: timothy.springer@childrens.harvard.edu).

ntegrins function by integrating the extracellular and intracellular environments in a bidirectional manner, with their extracellular domains binding to ligands while their cytoplasmic domains engage the cytoskeleton[1]. The integrin lymphocyte function-associated antigen-1 (LFA-1) is composed of the αL and β2 subunits. LFA-1 is expressed on all leukocyte subsets and binds specifically to the intercellular adhesion molecules (ICAMs). Their interactions mediate antigen-specific and innate immune cell interactions, firm adhesion, transendothelial migration of leukocytes in diapedesis and migration in tissues[2,3].

Integrins have three distinct overall conformations: bent with a closed headpiece, extended with a closed headpiece, and extended with an open headpiece (Fig. 1a). Headpiece opening is intimately associated with rearrangements at the ligand binding site and converts integrins to their high affinity, extended-open active conformation[4,5]. Integrins have long been known to mediate transmembrane force transmission[6,7], and must be connected to the actin cytoskeleton to achieve this. In focal adhesions, which are widely studied because of their highly organized structures, talin and vinculin make up the force transduction layer by linking actin filaments to integrins[8,9]. However, whether force itself could directly regulate integrin activity is still an open question.

The actin cytoskeleton serves as the ideal candidate for coordinating the multiple molecules required for directed cell migration. If actin dynamics can orchestrate when and where integrins and associated downstream partners need to be activated, it obviates the need for highly coordinated regulation of multiple pathways that would separately activate integrins and the actin cytoskeleton. We have previously shown that when LFA-1 is activated, the cytoplasmic domains separate[10], and that a high-affinity state can only be reached when LFA-1 engages immobilized ICAM-1, for example, ICAM-1 on a substrate or cell surface[4]. Cells apply mechanical force to beads bearing integrin ligands[11] and when migrating on ICAM-1 (ref. 12). High-affinity integrins have been localized underneath the midbody or front of migrating T cells[13,14]. However, the relationship between integrin affinity and force application to the substrate is unclear. Still, these observations are consistent with our current cytoskeletal force model for integrin activation (Fig. 1b), which supposes that force acts as an effector to stabilize the extended-open, active state of LFA-1 (ref. 15). However, this has not been confirmed or measured in migrating cells.

Although it has long been known that integrins can couple extracellular ligands to the actin cytoskeleton, the functional relationship between ligand, integrin and actin has not been fully characterized during cell migration. We do not know whether applied force on integrins selectively traverses the β-subunit as predicted by earlier models. The spatial distribution of force application to integrins on a migrating cell has not been measured, nor have the dynamics of force application to integrins and how this force is coupled to actin retrograde flow. Here to address these longstanding questions, we have developed fluorescent tension-sensing integrins. Analysis of the intracellular, nanometre-scale readout of intra-integrin tension that these sensors provide reveals supporting evidence of actin-dependent physical force in directly regulating integrin activation during cell migration.

## Results

**Tensile force is transmitted through the integrin β-subunit.**
To test the cytoskeletal force model for integrin activation, we created tension-sensing integrins by inserting a Förster resonance energy transfer (FRET)-based tension sensor module at different positions along the cytoplasmic tails of the integrin αL

and β2 subunits (Fig. 1b–d). The insert points were chosen by analysis of conserved regions across integrin β cytoplasmic tails (CT; Supplementary Fig. 1a). The module, developed by Schwartz and colleagues to measure intramolecular forces in vinculin, consists of monomeric teal fluorescent protein (mTFP1) and monomeric Venus (mVenus) joined by a 40 amino-acid elastic linker[16]. Calibration with single molecule measurements demonstrated elastic linker elongation upon tensile force application, providing sensitivity in the range from 0–1 pN (high FRET) to 6 pN (low FRET) (Supplementary Fig. 2a). Such forces are in the ideal range of the 2 pN talin-dependent rupture force exerted by the cytoskeleton on integrin molecules[17].

We validated the principle of measuring tensile force in the integrin β-subunit cytoplasmic domain and tested whether the tension-sensor constructs remained functional using β2-TS3 and β2-cTerm tension sensors stably expressed in Jurkat T cells. The β2-TS3 sensor is N-terminal to most of the β2 CT, including talin and kindlin motifs, and should be able to respond to force if functional; β2-cTerm, being at the C-terminus, should not be force responsive (Fig. 1c,d). β2-cTerm also serves as an ideal internal control as it will most likely experience similar local environment as the other sensors, including potential forces and conformational changes unrelated to tension but which could affect the FRET pair. β2-TS3 cells imaged live with total internal reflection fluorescence (TIRF)-FRET migrated well and exhibited a wide range of highly dynamic FRET efficiency clusters—or patches—across the whole cell surface (Supplementary Movie 1); the range of corresponding intramolecular forces spanned from 0 to 6+ pN (Fig. 1e,f; Supplementary Fig. 2a–c). On the other hand, cells with the β2 control sensor (β2-cTerm), show high FRET (low force) across the whole-cell while still migrating normally (Supplementary Movie 1; Fig. 1e,f). Acceptor photo-bleaching experiments validated that our constructs exhibit true FRET, as can be seen by the increase in donor intensity after bleaching (Fig. 1g; Supplementary Fig. 3). Altogether, these results show that several integrins with inserted tension sensors still allow normal cell migration, and that we can measure intramolecular forces on integrins in migrating cells.

The cytoskeletal force model predicts that mechanochemical force should be selectively transduced through the β subunit rather than the α subunit (Fig. 1b). To test this, we employed the same design principle for αL as for β2, inserting the tension-sensor in three distinct positions (Fig. 1c), and expressed them in Jurkat T cells. Live FRET images of migrating cells show that none of the αL tension sensor constructs exhibit tension, or any other significant FRET changes; the signal appears to be very similar to the negative control, β2-cTerm (Fig. 1e). Because of the highly dynamic nature of the FRET signal and the rapidly changing morphodynamics of the migrating cells, we quantitatively compared the αL and β2 sensor data by averaging whole cell FRET levels in multiple migrating cells. Both of the αL-TS constructs and the control αL-cTerm and β2-cTerm constructs exhibited FRET efficiencies in a range corresponding to zero force, whereas the β2-TS3 sensor showed whole-cell FRET levels corresponding to an average force per integrin of $1.4 \pm 0.8$ pN (mean ± s.d, $N = 28$) (Fig. 1f). These results clearly show that in migrating cells, force acts on integrins through the β-subunit and not the α-subunit, in accordance with the cytoskeletal force model.

**β2 sensors placed N-terminal to NPxF motifs respond to force.**
We investigated the impact on expression and migration of sensors placed in different positions in the CT. All β2 constructs expressed well in transiently transfected 293T cells (lacking endogenous β2 integrins), with average levels above 50% of

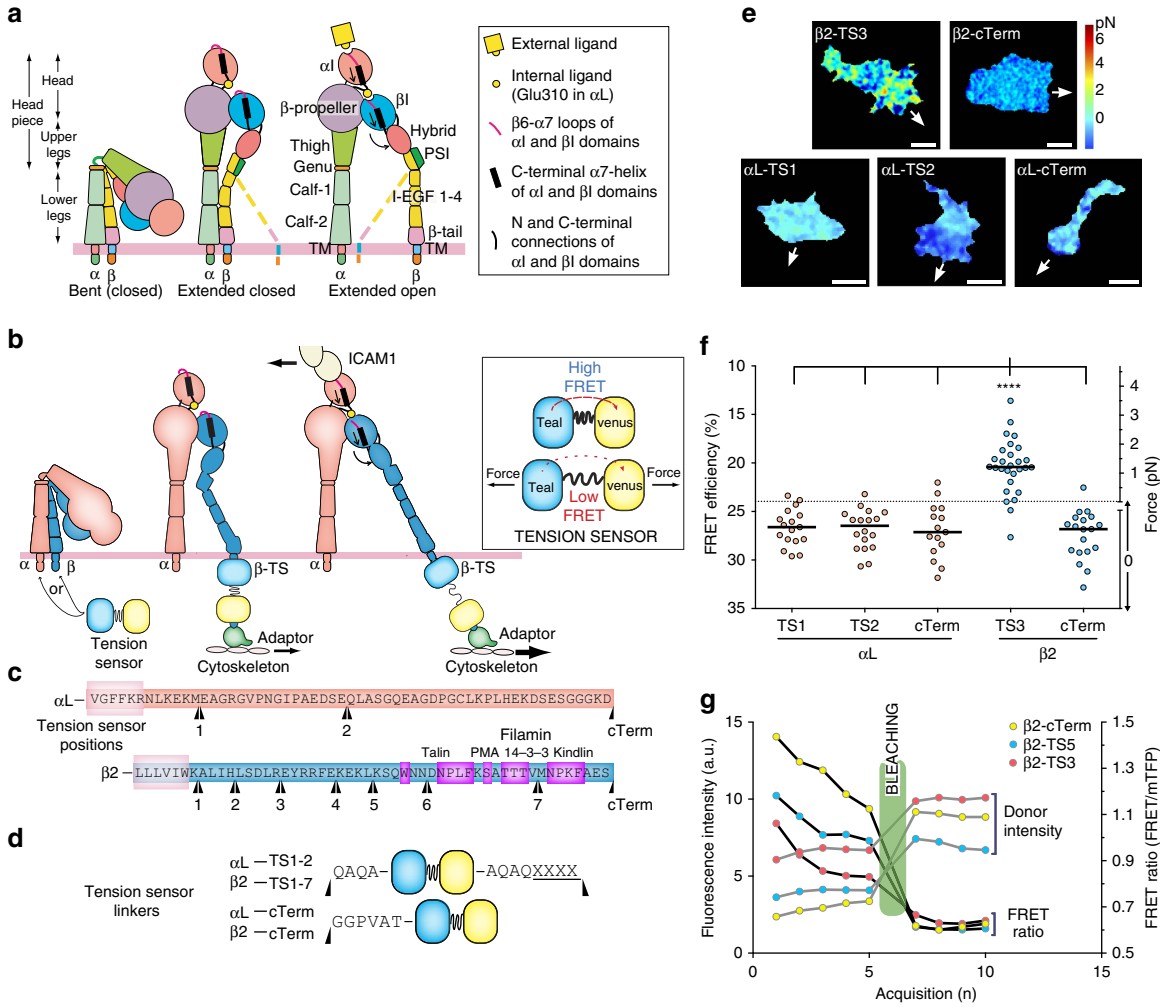

**Figure 1 | Tensile force is transmitted through the integrin β-subunit.** (**a**) Integrin structure and conformational states[60]. (**b**) Current model of actin-dependent integrin activation and illustration of the tension sensor module. The same conformations are shown as in **a**. The tension sensor (TS) consists of two FRET-compatible fluorescent proteins, monomeric teal (mTFP) and venus (mVenus), linked together with a repeating sequence that can be elongated by tensile force (GPGGA)$_8$. Black arrows represent force applied by the actin cytoskeleton and resisted by bound ligand[15]. (**c,d**) Tension sensor insertion positions where the preceding 4-residue sequences (XXXX) are being repeated except for the C-terminal constructs. Important interaction sites are highlighted in magenta. Linker details are in **d**. (**e**) Representative FRET images from movies of migrating Jurkat T cells expressing indicated tension sensor constructs. Arrow indicates direction of cell movement. Scale bar, 5 μm. (**f**) Whole-cell average FRET in migrating Jurkat T cells with 100 ng ml$^{-1}$ SDF-1α. Circles represent individual cells from three independent experiments with median shown as a line. ****: Kruskal–Wallis with Dunn's multiple comparison test of differences between β2-TS3 and all other TS had *P* values <0.0001. (**g**) Acceptor photobleaching of live Jurkat T cells. Curves depict donor (mTFP) intensity levels and FRET ratios before and after photobleaching of acceptor (mVenus). Representative curves out of 10 cells.

wild-type LFA-1 (Fig. 2a; Supplementary Fig. 4a). In comparison, the vinculin tension sensor expressed at 43% of wild-type levels[16]. Tension sensors centrally placed in the β2-CT, that is, TS3–TS5 (Fig. 1c) showed best expression, with close to wild-type LFA-1 levels (Fig. 2a).

To functionally compare the different β2 sensors we looked at their impact on LFA-1-mediated cell migration after being lentivirally transduced in Jurkat T cells. The β2 tension sensors constituted between 10 and 25% of all LFA-1 integrins on the cell surface of most cells (Supplementary Fig. 5f); the majority of the surface β2 still being wild-type allowed our probes to serve as passive observers of force, while minimally altering migratory function. We confirmed in a separate experiment that our sensor-bearing integrins were indeed functional by rescuing cell adhesion and cell migration in leukocyte adhesion deficiency patient cells deficient in β2 expression, although the rescue was less complete than with the wild-type β2 subunit (Supplementary Fig. 5d-e). Migratory ability was measured through live differential

interference contrast (DIC) imaging and automatic tracking of cells[18] migrating on ICAM-1 ± SDF-1α; as seen in studies with primary T cells[19], the addition of this chemokine[20] improved the overall migratory phenotype. Cells bearing the β2 sensor TS6 were the most affected, with elongated uropods suggesting inhibition of uropod detachment, while the β2-TS3, -TS5 and -cTerm were overall minimally affected as compared with regular Jurkat E6 controls (Fig. 2b–d; Supplementary Fig. 5a–c; Supplementary Movie 2). These sensors were therefore used in the majority of experiments in this study.

The cytoskeletal force model predicts that a fully active integrin couples to the underlying actin cytoskeleton via integrin β-subunit-associated adaptor molecules[21]. There are several potential molecules that could act as a molecular bridge from β2 CT to actin, including filamin[22,23], talin[24] and kindlin[25], all of which have binding sites in the β2 CT (Fig. 1c). By looking at force in the integrin at each β2 sensor position, it should be possible to pinpoint the general location of the force-generating

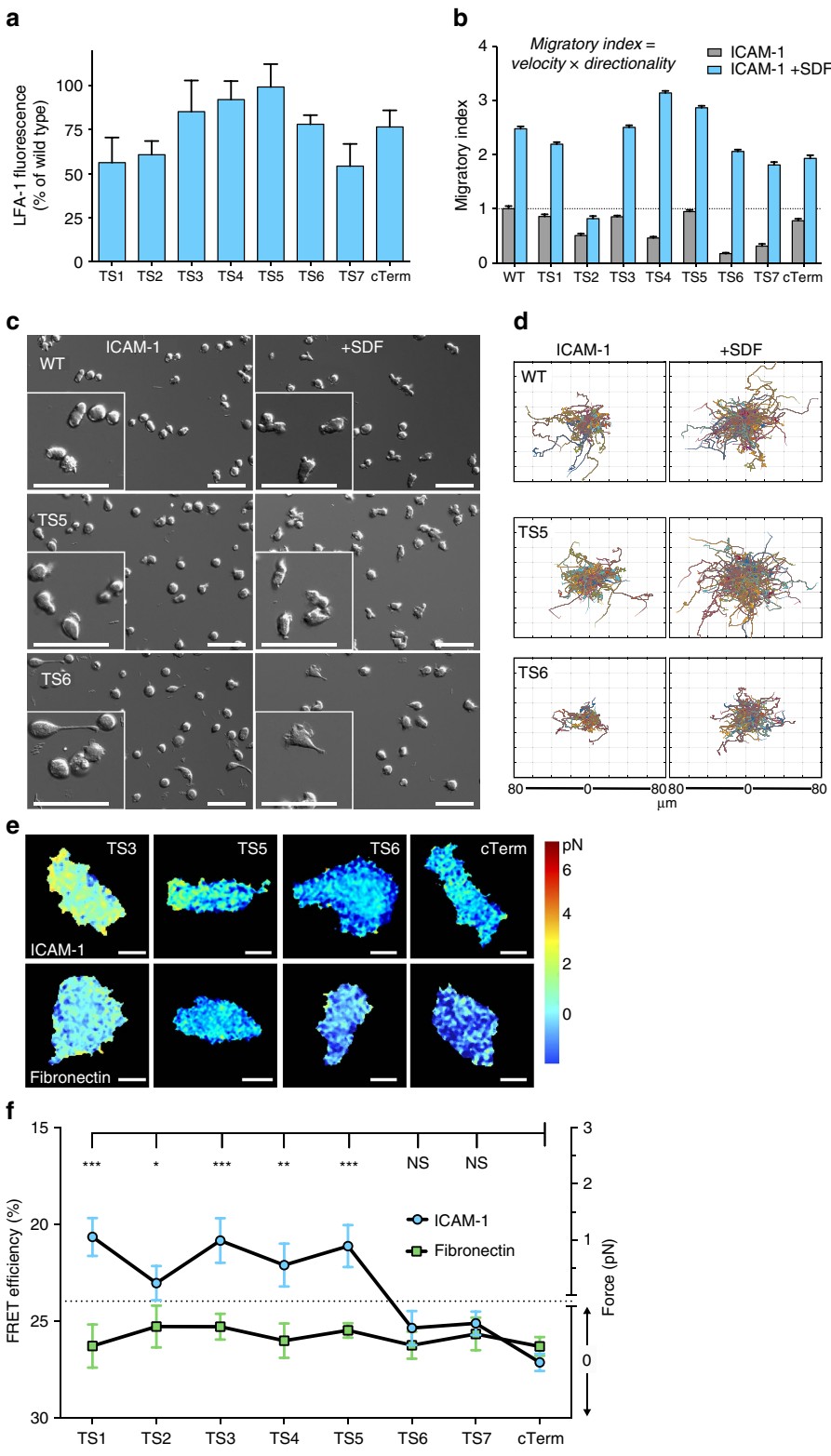

**Figure 2 | β2 tension sensors placed N-terminal to actin adaptor-binding sites respond to force.** (**a**) Transient expression of β2 tension sensors in 293T cells. Each sensor was transfected together with wild type αL. LFA-1 was quantitated by flow cytometry with the heterodimer-specific β2 monoclonal antibody TS1/18. Average values ± s.e.m. from three independent transfections. (**b**) Migration of Jurkat cells on 20 μg ml$^{-1}$ ICAM-1 with or without 100 ng ml$^{-1}$ SDF-1α. Each β2 construct was introduced by lentiviral transduction without addition of αL to allow pairing with wild type αL. WT is non-transduced Jurkat cells. From left to right: $N = 166/187$; 177/293; 102/121; 355/317; 215/350; 315/409; 186/233; 169/164; 348/159. Average values with ± s.e.m. are shown. Data are from three independent experiments. (**c**) Representative time-lapse DIC images of cells used for quantification of (**b**). Scale bars, 50 μm. (**d**) Representative cell tracks of cells used in (**b**) with all starting points shifted to the center. (**e**) Representative FRET images from fixed Jurkat T cells. Scale bar, 5 μm. Images are from 4 to 5 independent experiments. (**f**) Quantification of whole-cell FRET levels in fixed Jurkat T cells that were migrating on either ICAM-1 or fibronectin. Values are shown as mean + − s.e.m. Kruskal–Wallis with Dunn's multiple comparison test for β2-cTerm on ICAM-1 and the others yielded P-values: ***$P < 0.001$, **$P < 0.01$, *$P < 0.05$. No significant differences found when comparing cells on fibronectin.

adaptor sites. To more easily screen the different sensors, we measured intramolecular force on the integrin TS over whole cells, which were fixed during migration. Jurkat T cells express enough integrin α4β1 and α5β1 to support migration on fibronectin[26], and fibronectin was used as control when comparing FRET levels for different sensor positions (Fig. 2f). Sensors TS6, TS7 and cTerm showed comparable FRET intensity, which corresponds to zero force, on both ICAM-1 and fibronectin. TS6 and TS7 are inserted just N-terminal to the talin and kindlin binding sites, respectively, and might have disrupted binding to these sites, consistent with their effects on migration shown in Fig. 2b. In contrast, all β2 tension sensors placed N-terminal of TS6 showed higher force levels on ICAM-1 as compared with the cTerm control (Fig. 2e,f). This points to LFA-1 specific FRET activity for both of the sensors (TS3 and TS5) that were chosen for more detailed analysis, and demonstrates that these constructs in most cases both express and function well in a migrating cell.

**Integrin tension depends on interaction with specific ligand**. To investigate the dependence of integrin tension on the specific bound ligand, β2-TS5 and β2-cTerm Jurkat T cells were imaged during migration on ICAM-1, fibronectin, and anti-CD43 (for non-integrin based adhesion) substrates, and overall force was measured (Fig. 3a,b; Supplementary Movie 3). Similar to the results in fixed cells with multiple TS constructs (Fig. 2), force was exerted on TS5 on the specific ligand ICAM-1, but not on the control ligand fibronectin. FRET with TS5 on anti-CD43 was low and in the range corresponding to no force, but the FRET efficiency was higher than seen with cTerm on anti-CD43 or with TS5 or cTerm on fibronectin. This finding might suggest weak coupling between LFA-1 and CD43 during migration on anti-CD43 substrates.

To directly measure the vertical distribution of integrins in cells fixed during migration we collected TIRF z-stacks (with 25 or 50 nm step size) of TS integrins using the mVenus channel and using Lifeact-mCherry as a reference for cortical actin. As the focal plane in these stacks approaches the position of the average z-position of a diffraction limited signal, the images become more structured and less blurry, which can be measured by an increase in the pixel-to-pixel variance (Fig. 3c). Therefore, the relationship between z-position and normalized variance allows us to estimate the vertical (z) center of the fluorescence signal across entire cells (Fig. 3c,d), in a manner analogous to other precise axial-localization methods[27]. Although individual cells appeared to differ up to a few hundred nanometres in their absolute z position in the focal plane of the microscope, defining the z center of the integrin relative to the z center of actin in each cell provided consistent cell-to-cell results and a more precise estimate of integrin localization in various conditions. Relative distances between the average z-positions of actin and integrins in migrating cells were converted to absolute differences by correcting for chromatic aberration by analysing z-stacks of fluorescent 100 nm beads (Fig. 3c). These results show that for cells migrating on ICAM-1, LFA-1 integrins are located $45 \pm 50$ nm (mean $\pm$ s.d., $N = 56$) below the actin cortical layer (Fig. 3d). This is very close to the value of $60 \pm 20$ nm reported for αV integrins in focal adhesions using super-resolution imaging (Kanchanawong et al.[9]), suggesting that our estimates are valid for active integrins. Changing the substrate to either fibronectin or α-CD43 resulted in an upward shift of the integrin distribution, such that it was coincident with or above that of the actin layer (Fig. 3d). This suggests either that most of the inactive integrin is in an endosomal compartment, or that it is in plasma membrane regions that are on average higher above

the substrate than plasma membrane regions with active integrins that bring cortical actin close to the substrate. Overall, these results show that β2 integrin positioning and tension are dependent on interaction with a specific extracellular ligand.

**Integrin tension depends on both NPxF motifs**. Talin-1 has been shown to be critical for LFA-1 dependent T-cell migration[28], and kindlin-3 has been shown to be critical for spreading and adhesion of leukocytes on ICAM-1 (ref. 29). Both of these actin adaptors have roles in affinity regulation of LFA-1 (ref. 30). Hence, we introduced either a F754A or a F766A mutation in β2-TS5 to inhibit binding of talin[31] or kindlin[29] respectively. These NPxF-motif mutations also affect interactions with other proteins, such as filamin and 14-3-3 binding, but talin and kindlin are likely more associated with integrin activation[32]. The average force on TS5 was close to 1 pN on ICAM-1 substrates and was in the zero pN-range on anti-CD43 substrates, and independent replicates for these conditions in Fig. 4 gave very similar FRET values to those shown in Fig. 3. Strikingly, both the F754A mutation and the F766A alone were sufficient to reduce overall force on the integrin to the zero range (Fig. 4a,b; Supplementary Movie 4). Given the requirement for actin adaptors, we assessed whether the presence of polymerized actin itself is necessary for integrin force. We imaged migrating T cells acutely treated with a low-dose of cytochalasin D (250 nM) to stop actin polymerization in a way that would not immediately inhibit cell migration, and then continued to image as the drug started to function. Similar treatment has earlier been shown to increase the lateral mobility of LFA-1 in T cells[33]. Our results show a time-dependent reduction in force levels coupled with inhibition of cell migration, indicating that active actin polymerization is a requirement for LFA tensile force generation (Fig. 4c). Analysis of vertical localization of integrins with NPxF mutations shows a vertical upward shift of the population for both point mutations (Fig. 4d,e), akin to effects of removing the ligand (Fig. 3d), and indicating that most of these mutated integrins are not bound to the substrate. These results suggest that tension on the integrin is dependent on interactions with both talin and kindlin and that this mediates integrin ligand binding.

**Integrin tension correlates with active conformational state**. To address the question of how integrin conformational state correlates with force exertion, we examined the relationship between conformation-specific antibody staining and FRET in each pixel. The binding sites and the conformations the anti-bodies recognize are shown in Fig. 5a. The results are complicated by the co-expression of tension-reporting and wild-type LFA-1– necessary to draw conclusions on the NPxF point mutations in a migrating cell and the issue of antibody access to integrins in confined cell-substrate contact zones that bind to epitopes very close to the ICAM-1 binding site. The latter is particularly an issue for the m24 antibody specific for the high affinity integrin state[34]. For the TS1/18 antibody, which binds to integrins with a closed headpiece (Fig. 5a), we see a slight increase in whole-cell stain intensity (normalized MFI $\pm$ s.e.m.; $1.58 \pm 0.13$ and $1.79 \pm 0.14$) with the TS5 NPxF point mutations, indicating that more integrins are inactive in these cells relative to those expressing un-mutated TS5. We do not observe any pixel-by-pixel correlation between FRET ratios and TS1/18 fluorescence (Fig. 5b,c). For KIM127, which recognizes extended integrin conformations (Fig. 5a), pixel-by-pixel variations in antibody fluorescence intensity correlated inversely with TS5 FRET ratio, indicating a relationship between intra-LFA tension and the extended conformation (Fig. 5b,d). Furthermore, we saw no

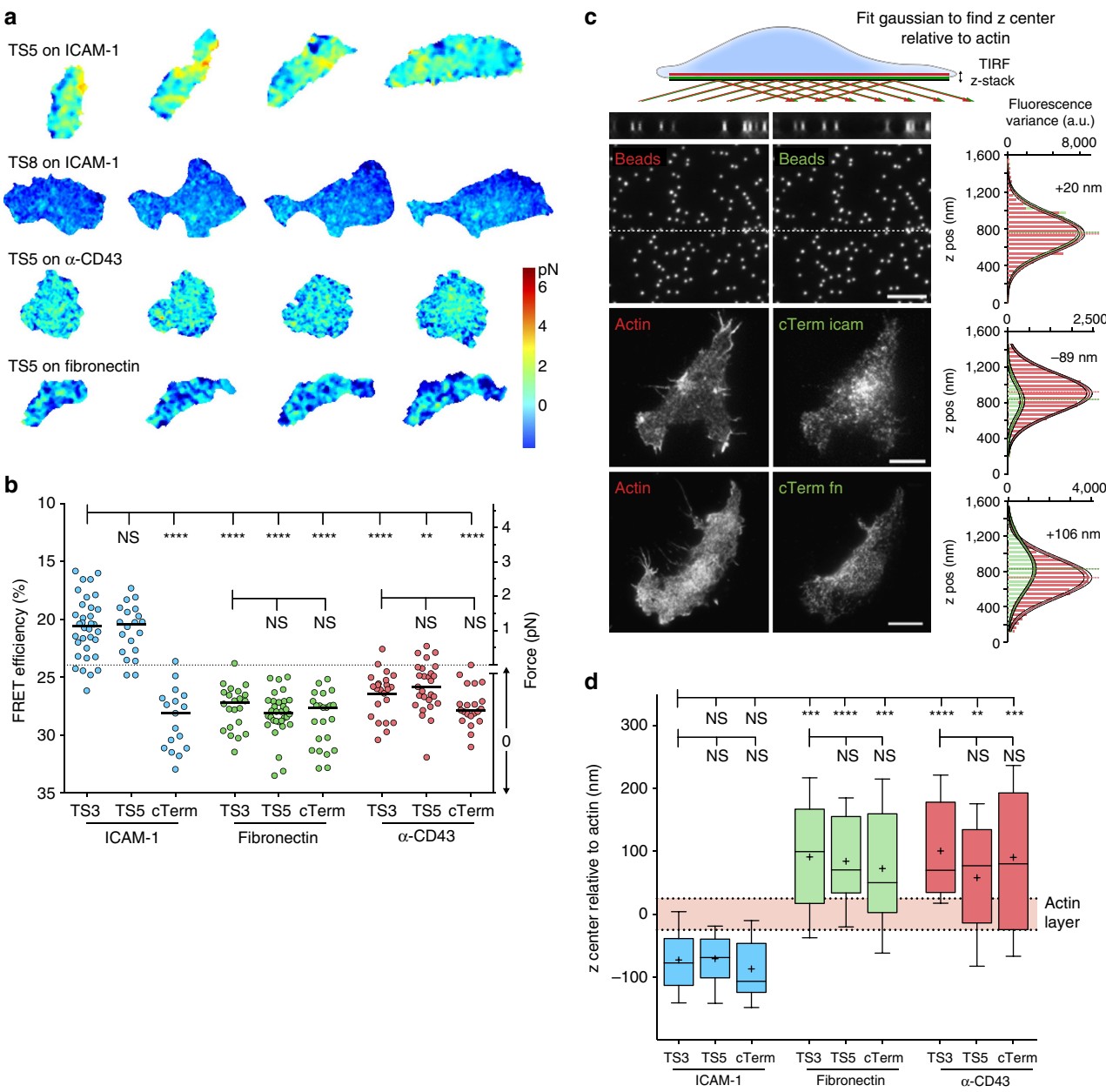

**Figure 3 | Integrin tension depends on interaction with specific ligand.** (**a**) Representative FRET images over 60 s of migrating Jurkat T cells expressing indicated tension sensor constructs. Images are corrected for photobleaching. Images are from three independent experiments. (**b**) Whole-cell FRET levels in migrating Jurkat T cells with 100 ng ml$^{-1}$ SDF-1α. Circles represent individual cells from three independent experiments with median shown as a line. Kruskal–Wallis with Dunn's multiple comparison test of differences between indicated conditions yielded P-values: ****P < 0.0001, ***P ≤ 0.001, **P ≤ 0.01. (**c**) Representative images of focal plane in TIRF z-stacks (50 nm step) of Lifeact-mCherry and TS-mVenus with distribution of fluorescence variance and Gaussian fits with indicated relative difference of the whole-cell distribution. The dotted lines across the submicron bead (100 nm) images are shown as side views at the top. Scale bar, 5 μm. (**d**) Relative differences of integrin z center to actin after chromatic aberration correction (N = 15,13,15,16,20,22,16,16,11). Box plots show the 5–95 percentile range (whiskers) of observations with median as line, mean as plus sign, and 25–75 percentile range boxed. Mean ± s.e.m. from left: −42.9 ± 12.4, −40.8 ± 11.4, −57.2 ± 12.7, 121.3 ± 24.0, 114.5 ± 17.5, 102.8 ± 20.33, 130.8 ± 20.3, 88.1 ± 23.7, 120.5 ± 32.2. Kruskal–Wallis with Dunn's multiple comparison test of differences between indicated conditions yielded P-values: ****P < 0.0001, ***P ≤ 0.001, **P ≤ 0.01.

correlation of KIM127 staining with FRET ratios of cTerm or TS5 NPxF site mutants. Moreover, we saw a similar, albeit weaker, inverse correlation of TS5 FRET with m24 fluorescent staining, which recognizes the extended-open conformation (Fig. 5b,e), and the correlations were significantly higher than for TS5 CT mutants or cTerm. These results show that LFA-1 extension and headpiece opening—and thus the active integrin conformation—appears to correlate with force application.

**Strong actin retrograde flow in conditions with low tension.** The cytoskeletal force model predicts that bulk actin retrograde flow should decrease as actin couples to the integrin β-subunit CT. To test this, we acquired high-speed images (1 frame per s) from migrating cells expressing Lifeact-mCherry and measured actin velocities using kymographs. Wild-type Jurkat cells showed slow actin retrograde flow during cell migration (− 17 nm s$^{-1}$), and in many cases even zero flow (Fig. 6a,b;

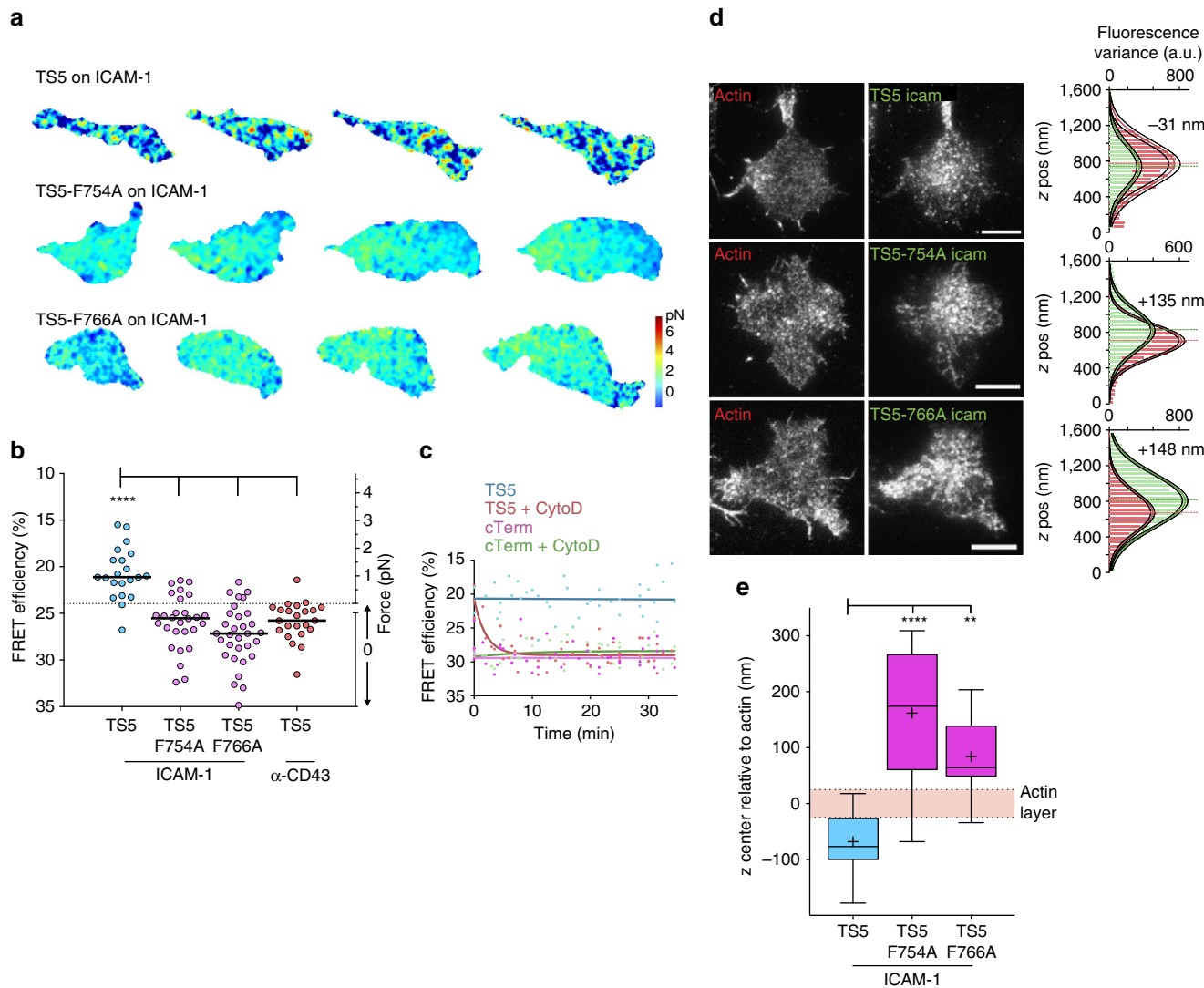

**Figure 4 | Integrin tension depends on both NPxF motifs. (a)** Representative FRET images taken from movies of migrating Jurkat T cells. Images are corrected for photobleaching. Images are from three independent experiments. **(b)** Whole-cell FRET levels in migrating Jurkat T cells with 100 ng ml$^{-1}$ SDF-1α. Individual cells from three independent experiments are shown as circles with median as line. Kruskal–Wallis with Dunn's multiple comparison test of differences between β2-TS5 and the other conditions all had P values < 0.0001. **(c)** Average FRET efficiency of migrating cells treated with 250 nM cytochalasin D at time zero. Each point indicates one cell, imaged only once to avoid complications from photobleaching. Data from two independent experiments are fit to single exponentials. **(d)** Representative images of focal plane in TIRF z-stacks (50 nm step) of Lifeact-mCherry and TS-mVenus with distribution of fluorescence variance and Gaussian fits with indicated relative difference of the whole-cell distribution. Scale bar, 5 μm. **(e)** Relative differences of integrin z center to actin after chromatic aberration correction (N = 13, 16 and 11). Box plots show the 5–95 percentile range (whiskers) of observations with median as line, mean as plus sign, and 25–75 percentile range boxed. Mean ± s.e.m. from left: − 68.0 ± 17.1, 161.8 ± 28.9, 84.1 ± 20.5. Kruskal–Wallis with Dunn's multiple comparison test of differences between indicated conditions yielded P values: P < 0.0001 and P = 0.0013.

Supplementary Movie 5). Cells that expressed β2-TS3, -5 and -cTerm had retrograde flow velocities of a similar range (− 13–33 nm s$^{-1}$). The TS5-F754A and F766A mutants show that loss of integrin linkage to actin (likely via talin and kindlin respectively) results in an increase of observable actin retrograde flow to − 114 and − 118 nm s$^{-1}$, respectively (Fig. 6a,b). These cells have a large portion of wild type β2 integrin present—which is necessary for wild-type migration—and most of the actin is likely still being engaged by those β2 integrins with intact actin adaptor binding regions. The movies represent some of the clearest examples of actin flow (Supplementary Movie 5), and in most cells it is only detectable after kymograph generation. Not surprisingly, the fastest actin flow was observed on a non-integrin substrate (α-CD43) (Fig. 6c), with average velocities of − 182–

204 nm s$^{-1}$), although cells on this substrate did not migrate well. These actin flow results are similar to those seen with $b2^{-/-}$ or $talin^{-/-}$ dendritic cells[35]. Force applied to β2-TS5 and the F754A and F766A mutants in actin flow experiments was comparable to that measured in Figs 2–4, as confirmed by capture of all channels in the first frame, followed by imaging in the actin channel at high speed (1 frame per s; Fig. 6d). Altogether, with quantification of force under identical conditions (Figs 3b and 4b), this shows that integrin tension is only seen under conditions where actin retrograde flow is low (Fig. 6e). The results further support the importance of kindlin and talin binding in allowing the force arising from actin polymerization and flow to be transduced through the β2-integrin CT in migrating T cells, decelerating this flow and converting it into force applied to the substrate.

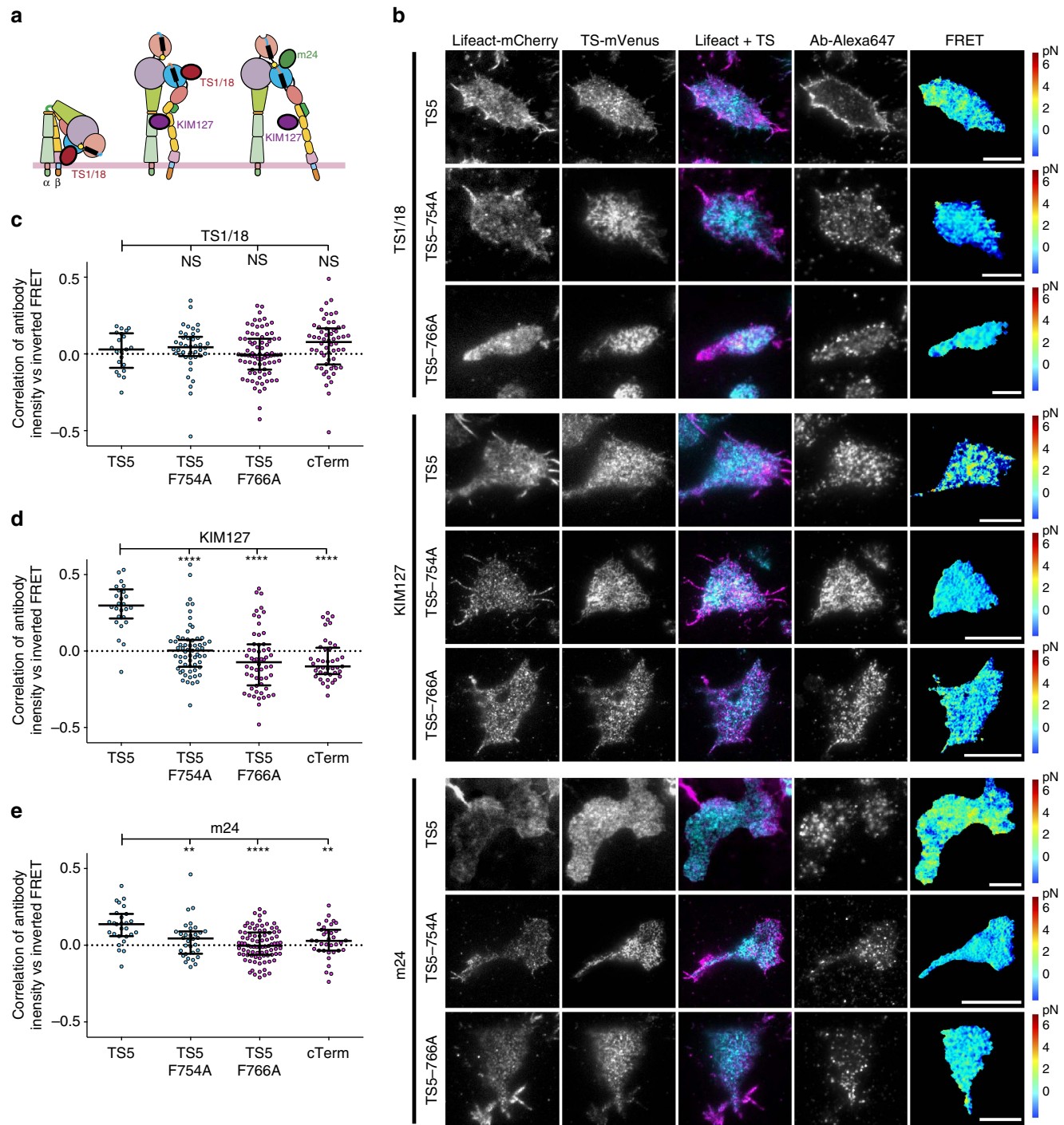

**Figure 5 | Integrin tension correlates with active conformational state. (a)** Integrin conformational states showing binding sites for antibodies used. **(b)** Representative TIRF images of migrating Jurkat T cells on ICAM-1 expressing indicated tension sensor constructs and stained with indicated antibodies. Scale bar, 5 μm. Images are from five independent experiments. **(c–e)** Correlation of antibody fluorescence intensity with inverted FRET ratio. Circles represent Pearson's correlation coefficient from individual cells from five independent experiments with median shown as a line. Mean correlation values from left, TS1/18: −0.017, 0.001, −0.059, −0.032; KIM127: −0.304, 0.061, 0.057, −0.010; m24: −0.127, −0.004, −0.025, −0.039. Kruskal–Wallis with Dunn's multiple comparison test of differences between β2-TS5 and other constructs were not significant for TS1/18, all P values < 0.0001 for KIM127, and for m24 the P values were $P = 0.0065$, $P < 0.0001$ and $P = 0.0062$.

**Protrusion can be powered by low force on integrins.** Our observations so far are in line with an actin-dependent force mechanism for integrin activation. To visualize and quantify the initial events occurring as the integrin is becoming fully activated, we used multichannel imaging to sequentially capture actin and FRET channels at high speeds to be able to correlate them.

We found that in many cases high integrin tension was localized to the leading edge of migrating cells and coupled with an increase in actin polymerization (Fig. 7a). Given the enormously variable morphology and low persistence of directionality in T-cell migration even during chemotaxis (data not shown), we applied computational morphodynamical and FRET pattern

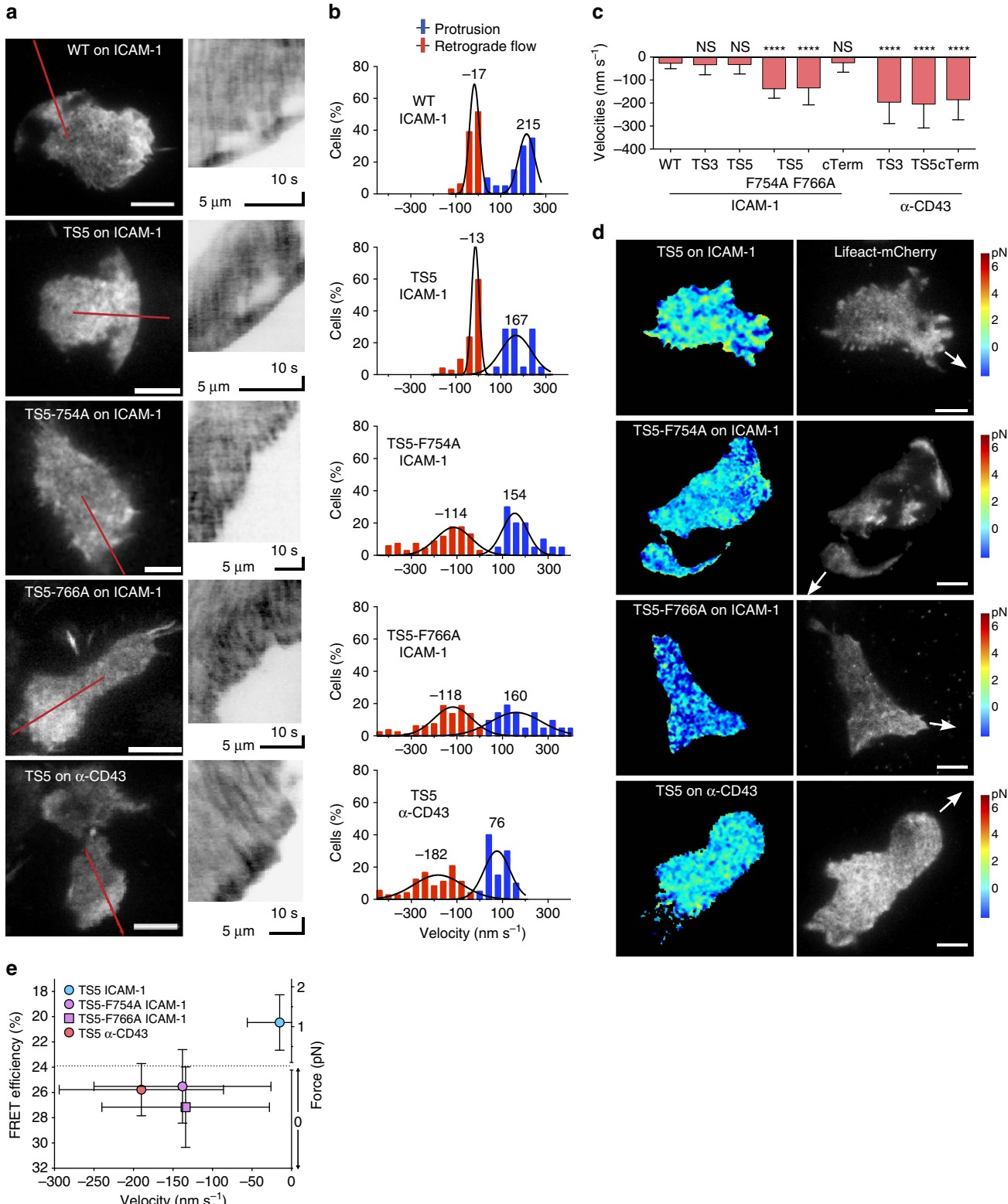

**Figure 6 | Strong actin retrograde flow in conditions with low integrin tension.** (**a**) TIRF images of Lifeact-mCherry in migrating Jurkat T cells with 100 ng ml$^{-1}$ SDF-1α. Red lines show regions from which kymographs (right) were generated. Images are from 3 to 6 independent experiments. Scale bar, 5 μm. (**b,c**) Actin retrograde flow and protrusion velocities from kymographs. Gaussian fits to the data are displayed ($N = 64, 55, 72, 68, 60, 45, 38, 72, 42$ kymographs analysed from 15–25 cells per condition). Values are shown as mean ± s.d. Kruskal–Wallis with Dunn's multiple comparison test of differences between WT cells and other conditions yielded $P$ values: ****$P < 0.0001$. (**d**) Representative FRET (colour) and actin (grey scale) images before start of high-speed actin movies quantified in **b**. The arrows indicate the direction of cell movement. Scale bar, 5 μm. Images are from three independent experiments. (**e**) FRET and force versus actin retrograde flow. FRET data from Figs 3b and 4b are shown with retrograde flow data from **d** as mean values ± s.d.

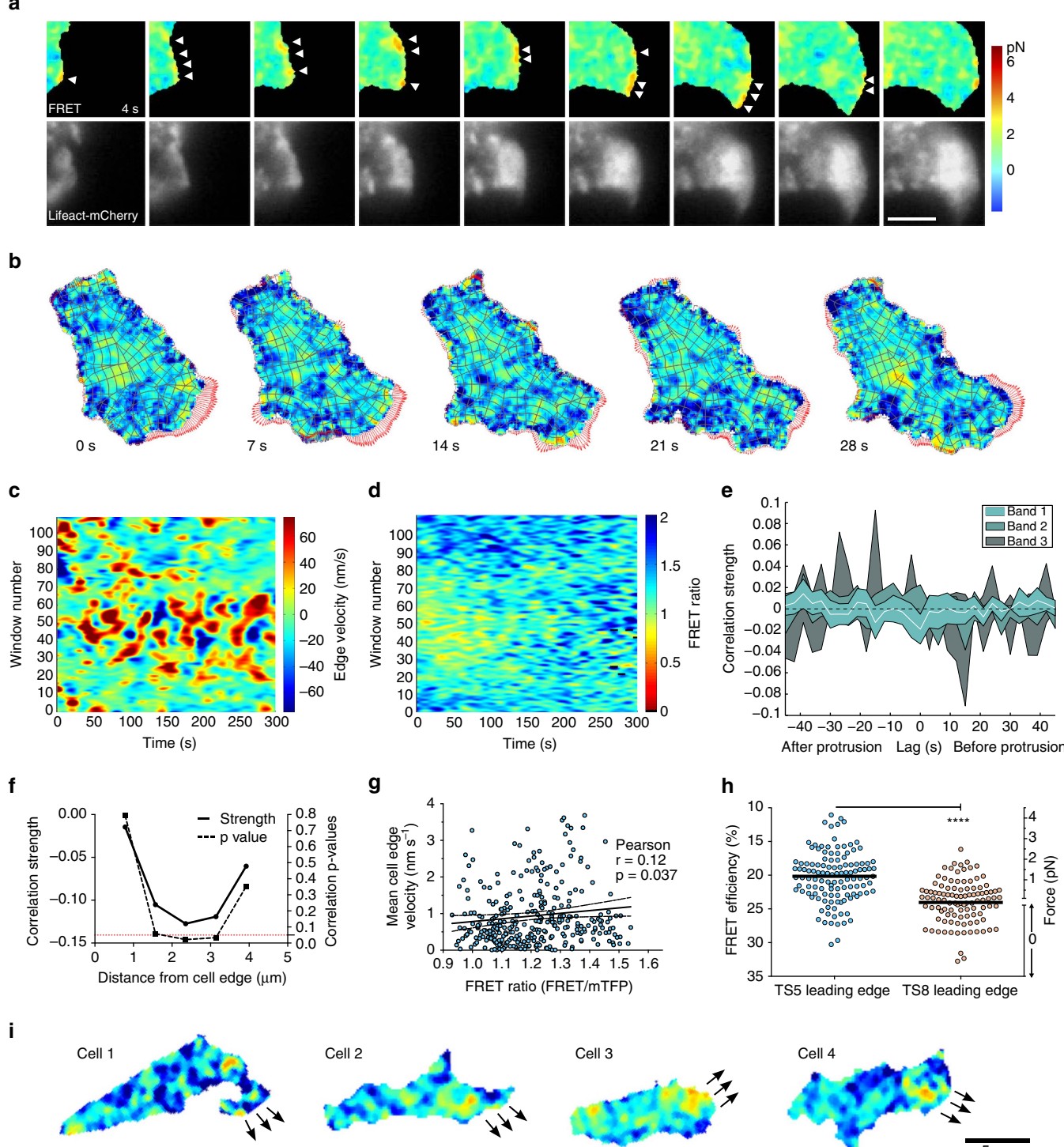

**Figure 7 | Protrusion can be powered by low force on integrins.** (**a**) Representative FRET and actin image series on migrating Jurkat T cell taken every 3 s (from >10 independent experiments). Arrowheads point to high force regions that correspond with protrusion in the following frame. Scale bar, 5 μm. (**b**) Image series from morphodynamics analysis. Each window (gray polygons) is tracked over time. Red arrows show edge velocity vectors. Force scale as in **a**. (**c,d**) Example of morphodynamics analysis of protrusion activity (**c**) and FRET (**d**). The windows are organized in bands along the cell periphery, with band 1 being all windows along the cell edge, band 2 being the row behind, and so on. Window 55 is near the front of the migrating cell and windows 1 and 110 are at the cell rear. (**e**) Correlation of instantaneous force versus protrusion over time in TS5 cells. In band 1 there is a weak but significant inverse correlation at zero lag. (**f**) Correlation of force vs protrusion localization. By averaging windows across the entire movie, a peak correlation value is found at 2.4 μm (band 3) from protruding edge. (**g**) Distribution of mean force vs mean cell edge velocity over time with Pearson correlation (*r* and *P* value shown) and linear regression plotted as solid line with 95% confidence interval shown as dashed lines. $N = 324$ protrusion events. (**h**) Leading edge quantification. The average force at the leading edge of migrating cells was measured to be 1.5 pN for TS5 cells ($N = 122$) and 0 pN for cTerm cells ($N = 105$). A two-tailed Mann–Whitney test yielded a $P$ value $< 0.0001$. (**i**) Representative images from migrating TS5 cells showing peak force localization behind the leading edge (from >10 independent experiments). Force scale as in **a**. Arrows indicate edge movement in next frame.

extraction analysis to generalize and quantify observations from multiple cells[36,37]. Each cell is divided into analysis windows (Fig. 7b–d, Supplementary Movie 6) in which local morphodynamics are related to actin or FRET signals at different distances from the cell edge. Cross-correlation of each window across movies from β2-TS5 cells showed a significant but weak instantaneous temporal correlation ( − 0.02) between force and protrusion at the cell edge (band 1), coinciding with zero time delay (Fig. 7e); temporal correlations for the other bands were not statistically significant. The weak instantaneous correlation could be due to noisy or under-sampled signals, or could highlight the heterogeneous nature of integrin force application during T-cell migration. Analysis with respect to integrin tensile force localization revealed a significant correlation ( − 0.13) between areas of high integrin tension and protrusive edge regions during cell migration (Fig. 7f); the strongest correlation was located on average ∼2.4 µm behind the cell edge. There also appeared to be a tendency for lower force as the cell edge velocity increased, which could be explained by decreased adhesion facilitating higher speed (Fig. 7g). Cells expressing the control cTerm construct exhibited very different patterns than the TS5 cells (Supplementary Fig. 6a,b), including a different distribution of FRET values (Supplementary Fig. 6c), but the overall cell edge velocities were not significantly different (Supplementary Fig. 6d). The above results suggest that a fairly low force is sufficient to power the protrusion of the leading edge, although sometimes the cells will use the leading edge as their main propulsion point with corresponding high force in those areas. The FRET levels we observed at the cell edge corresponded to average integrin tensile forces of 1.5 (Fig. 7h). On the basis of our results, typical force peak localization in a migrating T cell is exemplified by images from four different migrating cells (Fig. 7i). This demonstrates that while the relationship is noisy, there is a measureable relationship between intracellular integrin force distribution and local morphodynamic variability.

## Discussion

The mechanism of integrin activation has been investigated for many years and by many labs. While force from actin has been previously assumed to be responsible for activation, direct evidence was lacking, and the effect of force on integrins could easily have been mediated by intermediate biochemical or biophysical cues. Here, we provide evidence that for migrating T cells, actin-derived tension is required for LFA-1 activation, and that furthermore this activation requires the correct ligand and intact talin- and kindlin-associated binding motifs. Only when all these requirements were satisfied was LFA-1 able to bind to the ICAM-1 substrate, demonstrating that integrin activation is stabilized by cytoskeletal force. On the basis of these findings, we propose an updated model for actin-dependent integrin activation during cell migration (Fig. 8).

Critical to the results presented in this study is the functionality of the integrin tension sensor constructs. A range of different controls or experimental results support their functionality: acceptor photobleaching FRET validation experiments; surface expression in αLβ2-lacking cells and co-expression in αLβ2-containing cells; dominant negative effects on integrin function by some sensor constructs but not those used in this study; rescue of adhesion and migration in β2-deficient leukocytes; three-dimensional localization of integrin sensors reflecting that of regular β2-integrins; changing substrate yields consistent changes in both FRET, actin flow and ligand proximity; introducing point mutations in the NPxF motifs yields consistent changes in both FRET, actin flow and ligand proximity; confirmation of many results by two independent

β2 integrin sensor constructs; β2-cTerm, an ideal control for any cell-specific or assay-dependent effects on both FRET and integrin behaviour, is present across the experiments. These integrin tension sensors will likely serve as valuable tools for studies of other aspects of integrin activation and force transduction, and given that multiple constructs appear to work well, indicates that the design principles could be applied to create other β-integrin biosensors as well.

From studies on αXβ2 and αLβ2, we know that headpiece opening with hybrid domain swing-out is required for cell adhesion and high affinity for ligand[4,34]. This swing-out step was suggested by molecular dynamics simulations to be mediated by a lateral pulling force on the β-subunit cytoplasmic tail[15]. Necessary counter force comes from an immobilized ligand, which is a requirement for LFA-1 headpiece opening and adhesiveness[4]. So far, to our knowledge, there has been no successful reconstitution of inside-out signalling-induced integrin headpiece opening in vitro. Extension of integrins in lipid nanodiscs can be induced by talin alone (22% extension[38]); however, the integrins all had a closed headpiece and thus lacked high affinity for ligand[5]. Thus the question of whether force is required for integrin activation—in the sense of stabilizing integrin binding to ligand—has heretofore remained unanswered.

Crystal and EM structures of the β2 integrins αXβ2 and αLβ2 allow us to describe the pathway through which tensile force flows[34,39–41]. Tensile force is exerted though the αL-subunit αI domain from its ligand-binding metal ion dependent adhesion site (MIDAS) to its internal ligand, crosses to the β2-subunit where the βI MIDAS binds the αI internal ligand, is transmitted from the βI domain to the hybrid domain, and is then transmitted down the β leg and through the plasma membrane to the cytoplasmic tail, which in turn connects through adaptors to the actin cytoskeleton (Fig. 1a,b). Force passing through the junction between the βI and hybrid domains favours the open conformation of the integrin headpiece, with its longer path for force transmission, which stabilizes the high affinity state by $\sim F \cdot \Delta x$ (ref. 42). Similarly, force favours integrin extension owing to the much greater distance between the αI domain and the β-subunit TM domain in the extended than bent conformations (Fig. 8). Overall, the findings presented here confirm the predicted force-bearing axis through the integrin β-subunit to its cytoplasmic domain. Furthermore, the previous structural studies and predictions fit our demonstration here that active integrins bear force, and strongly suggest that the active state interrogated here is the extended, open integrin conformation. This conclusion was further strengthened using conformation-specific mAbs. By showing that kindlin- and talin-binding NPxF motifs are simultaneously required for force transduction, we have not only validated this molecular biophysical model in actively migrating cells, but provided new insight into the force transduction pathway and its key players.

It is essential for a cell to be able to control when and where it will move in a certain direction. Multiple signalling pathways, especially those that regulate actin polymerization, actin reorganization and integrin activation, must be coordinated for migration to take place. We demonstrate a need for intracellular actin-generated force to facilitate efficient integrin-ligand engagement on the extracellular side. By connecting extracellular ligand binding to actin engagement in a positive-feedback loop, the cell can control its migration by coordinating actin activity. Since propulsive force is a prerequisite for migration, this feedback makes that force locally adaptive, allowing the cell to fine-tune where it needs traction—provided by the integrins—which then become active as force is applied by the cytoskeleton. Alternative mechanisms for a force–activation relationship that require biochemical intermediates would also imply complex centralized

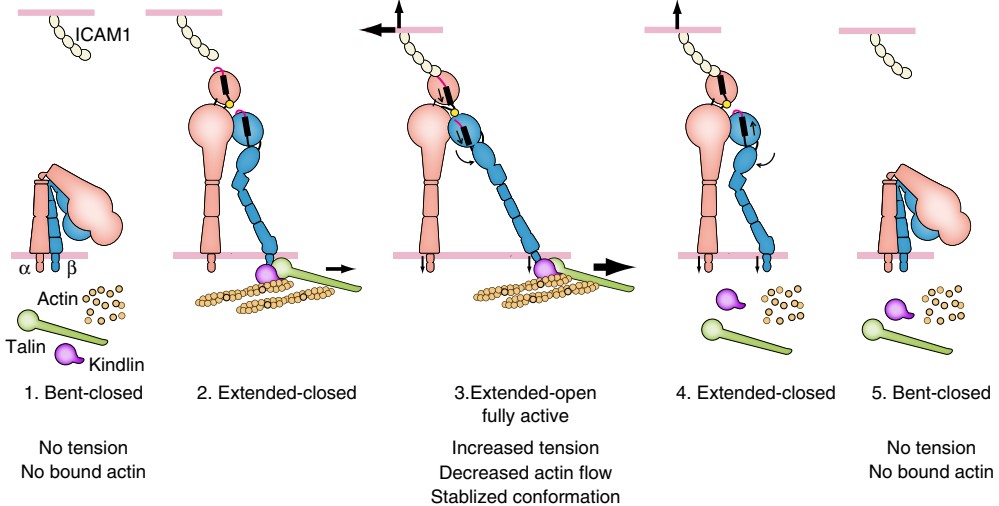

**Figure 8 | Model for actin-dependent integrin activation in cell migration.** (1) Inactive LFA-1 is neither bound to ICAM-1 on the extracellular side nor linked to actin on the cytoplasmic side. (2) Talin and kindlin link actin to the cytoplasmic tail of the β2 subunit. (3) When LFA-1 binds ICAM-1 and the actin cytoskeleton simultaneously, tensile force exerted across LFA-1 stabilizes the high-affinity conformation of the αI domain bound through its internal ligand to the βI domain with the open headpiece conformation and the hybrid domain swung out. (4–5) When the intracellular link to actin is lost, the open headpiece conformation is no longer stabilized, and the integrin returns to its lower energy, bent-closed conformation. The stability of the states is likely more relevant than the order of binding and conformational change. Previous work has shown that LFA-1 has 1,000-fold lower affinity for ICAM-1 in the bent-closed and extended-closed conformations than in the extended-open conformation[4,34]. The bent-closed conformation is more populated on the cell surface than the other states under resting conditions[4,39], but rapid kinetic exchange between the three conformational states at thermal equilibrium likely results in significant population of extended-closed and extended-open states in basal conditions[41]. Under basal conditions, individual integrin molecules may exchange thousands of times per second between conformational states, enabling them to sample CT binding to adapters and ectodomain binding to ICAM-1. These processes would enable a pre-existing extended integrin conformation to bind ICAM-1 and actin adaptors at the same time. Once pulling on the integrin by the cytoskeleton meets resistance from ICAM-1 anchored on the surface of another cell, tensile force exerted on the integrin between the ligand anchor point and polymerized actin stabilizes the open, high-affinity integrin conformation[41,42]. The molecules in the model are not drawn to scale.

regulatory networks to control integrin activation. In contrast, direct regulation by tension reinforces the idea that localized actin polymerization can act as a coordinator for activation of the complex machinery required for moving a eukaryotic cell in a certain direction, simplifying the necessary regulation and relaxing the requirement for centralized control of the multiple pathways involved.

## Methods

**Reagents.** Human SDF1-α was from R&D Systems. Wild-type soluble ICAM-1 (D1–D5) was expressed in 293T cells and purified on NiNTA agarose[4]. OxyFluor was from Fisher. Cytochalasin D was from Santa Cruz. Glass-bottom dishes and plates were from Mattek. Leibovitz's L-15 medium was from Life Technologies. The reagents for the lentiviral Gateway system were from Invitrogen. Monoclonal antibodies used were TS1/18, KIM127 and m24, see also antibody staining section.

**Cells.** Jurkat T cells (clone E6.1 from ATCC) were cultured in RPM1-1640 medium with 10% FBS in 5% $CO_2$. HEK-293 (ATCC) and HEK-293T cells (ATCC) were cultured in DMEM medium with 10% FBS in 10% $CO_2$. LAD patient cells were from a previously described patient cell line[43,44] (patient 2) and cultured in RPMI-1640 medium with 20% FBS, 50 μM 2-mercaptoethanol in 5% $CO_2$. The cell lines are not found in the NCBI Biosample database for commonly misidentified cell lines, and were not authenticated, but were tested for mycoplasma contamination.

**Constructs.** We compared integrin β CT sequences (Supplementary Fig. 1) to find non-conserved regions for sensor insertion. A four-residue relatively non-perturbing sequence with helical propensity, AQAQ, was used on each side of the tension sensor to minimize perturbation of surrounding CT structure (Fig. 1d). To further minimize disruption of potential CT binding sites on either side of the insertion, the four-residue CT sequence preceding the insertion point was repeated after the tension sensor module (Fig. 1c,d). This design was successfully piloted on β2-TS3 and then applied to all other inserts. Given the conserved CT sequence among the integrin β1, β2, β3, β5, β6 and β7-subunits, we believe that our design can be applied to make tension sensors for all 22 integrins with these β-subunits, enabling a wide range of force-dependent cellular processes to be explored. For the C-terminal sensors, a previously described successful C-terminal linker used with

integrins was employed[10] (Fig. 1d). Integrin tension sensor constructs were made using overlap PCR and inserted into pcDNA3.1( − ). The tension sensor module cDNA, encoding mTFP1-(GPGGA)₈-mVenus, was provided by Martin Schwartz[16]. Lifeact-mCherry was used to visualize f-actin[45].

**Lentiviral transduction of cells.** The Gateway system from Invitrogen was used to create lentiviral constructs. The integrin constructs were inserted into pLX302 and lifeact-mCherry was inserted into pLX304. Virus was produced in 293T cells by co-transfecting the lentiviral plasmids with psPAX2 and CMV-VSV-G. Virus in supernatants was concentrated using Lenti-X. Jurkat cells were transduced and selected using 3 μg ml⁻¹ puromycin (pLX302) or 1 μg ml⁻¹ blasticidin (pLX304). Jurkat cells expressed both endogenous and tension-sensing LFA-1 (Supplementary Fig. 5f), with the latter acting as reporters of integrin tension.

**Live imaging.** For the majority of live experiments, MatTek glass-bottom dishes or plates were adsorbed overnight at 4 °C with 10 − 20 μg ml⁻¹ fibronectin or ICAM-1 in carbonate buffer (pH 9.6), followed by blocking at 37 °C with 1% BSA in L-15 medium for 30 − 60 min, and washing with base imaging media consisting of L-15 supplemented with 2 mg ml⁻¹ glucose. Cells were suspended in base medium supplemented with lactate (10 mM), OxyFluor (1:100), and 100 ng ml⁻¹ SDF1-α. For some experiments, SDF1-α-infused agarose were placed as droplets or bars to induce a chemotactic response instead of a chemokinetic response. In all cases, cells were added to the dish or well on the microscope held at 37 °C and allowed to settle for 5–10 min. TIRF illumination was adjusted for each individual sample. Only samples with clearly migrating cells were used. The low fluorescence intensity inherent in FRET imaging with fluorescent proteins requires high laser power which in turn results in strong photobleaching (Supplementary Fig. 7). In single-chain FRET imaging, only the donor is excited by the laser, and it is excited twice, once to measure donor emission (TFP emission channel) and once to measure acceptor emission (Venus emission channel, that is, the FRET channel). Photodamage of the acceptor comes only from FRET, and is measured as a decrease in the FRET ratio (FRET emission/TFP emission). Such photobleaching effects on FRET ratios can be corrected for, and we have applied such correction to all movies (Supplementary Fig. 7, Supplementary Movie 7). However, to further ensure that no artifacts are introduced by correction, all force quantification reported here is based on the first frame of each movie. To minimize photobleaching, cells were typically located using actin fluorescence or DIC. Acquisition of images was then every 3 to 20 s.

**Fixation of samples.** For fixed cell analysis, cells were prepared as for live imaging, allowed to migrate at 37 °C for 20–30 min, and fixed by addition of an equal volume of paraformaldehyde to a final concentration of 3% for 10 min at 37 °C. After washing with L-15 medium, cells were imaged as for live samples. Fixation for antibody staining was done using the pH-shift method; see also antibody staining section.

**Image acquisition.** All tension sensor images were collected on a Nikon Ti inverted microscope equipped with a motorized TIRF illuminator and × 100 Plan Apo NA 1.49 objective lens and the Perfect Focus System for continuous maintenance of focus. The microscope was enclosed in a custom-built 37 °C incubation chamber. A Spectral Applied Research laser merge module with acousto-optical tunable filtre (AOTF) controlled solid-state lasers. A Prior Proscan III controlled filtres, shutters and stage. A triple band pass dichroic mirror (Chroma # 53055) was used for all acquisitions. mTFP1 fluorescence was excited with a 40 mW 446 nm laser and collected with a 500/40 emission filtre. mVenus fluorescence was excited with a 50 mW 515 nm laser and collected with a 535/30 emission filtre. mCherry fluorescence was excited with a 50 mW 561 nm laser and collected with a 636/60 emission filtre. For FRET channel images, the 446 laser was used in conjunction with the 535/30 emission filtre. The TIRF angles were set the same way for every experiment. Images were acquired with a Hamamatsu ImagEM C-9100-13 EM-CCD camera controlled with MetaMorph software. EM gain was kept constant at 200. The exposure times were 500 ms for mTFP1 and mVenus images. For lifeact-mCherry, the exposure times were between 100 and 200 ms. Image sets were collected every 3–20 s, depending on the experiment.

All cell migration images were collected on a Nikon Ti inverted microscope with a × 20 Plan Apo NA 0.95 objective lens with DIC optics in place and the Perfect Focus System for continuous maintenance of focus. The microscope was enclosed in a custom-built 37 °C incubation chamber. A Prior Proscan III controlled the stage and a Prior Brightfield LED illuminator was used. Images were acquired with a Hamamatsu ORCA-ER cooled CCD camera controlled with MetaMorph software. Images were collected every 30 s from multiple different stage positions.

**FRET image processing.** Image processing was mainly carried out using MATLAB 2014b. Functions handling all steps of the image processing were developed based on previous published algorithms[37,46]. Stepwise, images are imported from the original files and sorted into channels; all metadata is extracted and saved; dark current camera noise is corrected for by using images acquired each session; shading or 'flat field' correction based on empty fields from each sample is applied to correct for uneven illumination patterns; a background mask is generated by thresholding at a value 3 standard deviations above background, where the background intensity distribution is estimated by fitting the 'left half' of a Gaussian function (the portion below its mean) to the left shoulder of the image intensity histogram. This mask is then used to find and subtract the average background intensity on a frame-by-frame basis; for all FRET calculations the data is pre-filtered with a 3 × 3 pixel Gaussian filtre; since the tension sensor module is a single-chain construct, FRET can be calculated by dividing FRET channel (donor excitation with acceptor emission) with mTFP channel (donor excitation with donor emission); to minimize artifacts from division of small integers, only pixels that have a value above three times the background standard deviation of the current frame were used; FRET data was visualized using an inverted heat map. Photobleach correction was performed similarly as described[46] by fitting a single or double exponential to the ratio of average FRET/donor signal in each cell over the time-course. This exponential fit was then used to adjust the FRET signal over the time-course in relation to the initial time point. MATLAB functions for dark current correction, shading correction, background segmentation, background correction, FRET calculations and photobleach correction are available upon request.

The resulting FRET ratio ($R$) is only comparable for data with the same microscope and settings, whereas FRET efficiency ($E$) is comparable between systems. To estimate force ($f$) we use FRET efficiency measurements made by the Schwartz group for the tension sensor module (Supplementary Fig. 2a). Fitting their data to a fourth-order polynomial, we determined the relationship:

$$E(f) = 23.96 - 0.9686f - 1.391f^2 + 0.2212f^3 - 0.009531f^4 \quad (1)$$

To calculate the efficiency of our system we collected the data using constructs with known efficiencies for the same fluorescent protein pair, mTFP-5aa-mVenus ($E = 55\%$) and mTFP-TRAF-mVenus ($E = 11\%$ (ref. 47), Supplementary Fig. 2b). Given the Förster distance of our FRET pair, ∼6.0 nm, the relationship between FRET efficiency and ratio within the range ($E = 11$ to $55\%$) is approximately linear[48], so we used linear regression on the measured ratios in Supplementary Fig. 2b to find the equation:

$$E(R) = 27.363R - 11.219 \quad (2)$$

Force is estimated for FRET ratios measured in our system by numerically solving for $f$ in the equality of equations (1) and (2) at 1,000 points along the curve from equation (1) (Supplementary Fig. 2c). By fitting that data to a fourth-order polynomial, we determined the relationship:

$$R(f) = 1.286 - 0.03539f - 0.05083f^2 + 0.008086f^3 - 0.0003483f^4 \quad (3)$$

No simple analytical solution was found for expressing $f$ as the dependent variable in either equations (1) or (3).

**Image analysis.** All processed images for FRET analysis were segmented using either the actin channel or the combined TFP and FRET channels to identify and track individual cells. For quantification, an individual background noise estimation was applied (by fitting the portion of a Gaussian function below its mean to the left shoulder of the image intensity histogram) and only pixels with intensities at least three s.d. above that level were included as cellular segments. For FRET efficiency quantification, only the first frame was used unless indicated, to ensure no potential artifacts were introduced by photobleach correction.

Morphodynamic and windowing analysis was performed as described previously, with windows of 5 × 5 pixels using the 'constant number' propagation method[36,37,49]. For morphodynamics analysis, automatic cell selection was used; cells had to be segmented for at least 10 consecutive frames, without touching the frame edge or other cells, and had to have a signal-to-noise ratio (calculated as mean intensity above background divided by the background standard deviation) of at least five in their protrusive regions, which were defined as having a positive difference between frames for at least five frames and a total pixel increase over all frames in the protrusion of >1,000.

For migration analysis, a modified version of CellTracker[18] was used, with added visualization and statistics options. For actin flow measurements, manual line selections in Fiji were used together with the Kymograph plugin[50]. To measure the velocity of actin flow, the manual line selections were drawn perpendicular to the direction of migration and the quantification was done by taking the slope of linear kymograph features at the outermost section of the cells. At least 20 cells were analysed for each condition, and multiple measurements per cell were taken. To verify that this estimate could be reproduced by a different method, we also analysed a subset of data with a local gradient-based analysis method[51], where the outermost section just behind the cell edge of the kymographs were analysed. This yielded very similar results (data not shown).

**Cell segmentation.** Since multiple cells could be present in the same image and at different intensities, the background masking described above was used to initialize cell segmentation independently for each cell. Either active contour segmentation or intensity distribution-based threshold segmentation was used to produce an initial cell mask. Mathematical morphology (a closure operation with a radius of 1 pixel, small object removal, and filling of holes) was applied to further refine these masks, producing accurate cell outlines for morphodynamic analysis and whole-cell FRET calculations. To identify leading edges for the quantification in Fig. 6, edge masks were first created by taking the difference of our cell masks before and after a 10-pixel erosion. Protrusions were identified by finding mask increases between frames, checking for connectivity in the time dimension, a minimum empirically determined size (typically space-time volume of 1,000), and lasting at least 3 (for leading edge analysis) or 5 (for morphodynamics analysis) consecutive frames. Finally, these protrusion masks were combined with the previously generated edge masks to find leading edge regions.

**Acceptor photobleaching.** For acceptor photobleaching experiments five images were acquired before a 60 s 515 nm bleaching at 100% laser power, and then all channels were acquired again (Fig. 1g; Supplementary Fig. 3). As additional control of the system we imaged single-fluorophore-expressing cells as well (Supplementary Fig. 2d). There are reports that acceptor fluorescent proteins can photoconvert during acceptor photobleaching experiments[52], including mVenus used in this study[53]. This appears to be a concentration-dependent effect that is only present at high fluorophore concentrations, and is reduced for 1:1 stoichiometry of donor:acceptor and is 10-fold lower with mVenus compared with YFP[54]. Importantly, this will not be a factor at all for our FRET measurements as we only imaged using double donor excitation. Also, we did not use acceptor photobleaching to quantify FRET efficiency, instead using the well characterized mTFP-5aa-mVenus and mTFP-TRAF-mVenus constructs[47] for that purpose.

**Antibody staining and analysis.** Monoclonal antibodies used were TS1/18, KIM127 and m24 and they have been previously characterized[55–57]. Migrating cells were fixed using pH shift fixation[14], which constitutes an initial 3% PFA fixation in Pipes buffer (pH 6.8) for 5 min and a short wash with PBS immediately followed by 3% PFA fixation in $NaB_4O_7$ (pH 11) for 10 min. The samples were blocked with PBS with 5% goat serum and 100 mM glycine and permeabilized with 0.1% Triton X-100. The primary antibodies were incubated at 10 µg ml$^{-1}$ (TS1/18 and KIM127) or 30 µg ml$^{-1}$ (m24) overnight at 4C and goat anti-mouse Alexa647-Fabs was incubated for 1 h at 1:500 dilution. Cells were segmented using the mVenus channel. Pixels within the cell mask and more than 4 standard deviations above background in both the mTFP and FRET channel were correlated with corresponding pixels in the antibody channel using the MATLAB function corr2 from the Image Processing Toolbox, which is based on the Pearson correlation of 2D arrays.

**Z-stack analysis.** TIRF z-stacks were collected using either a 25 or 50 nm step size. The z-stack range included regions clearly out of focus both above and below the regular focus. Chromatic aberration was determined by acquiring z-stacks of beads for each session, and was between 20 and 30 nm. An ROI was set to cover each cell analysed, and the normalized fluorescence variance[58,59] in each plane was determined, with out of focus images having the lowest variance. The variance values were fit to a Gaussian function to estimate the vertical z center of the fluorescence distribution, and the integrin z center was compared to the actin z center of the same cell to normalize for cell-to-cell variation in vertical position on the substrate. No significant difference was found in actin–integrin relative positioning when comparing whole-cell analysis with front region (Supplementary Fig. 8).

**Flow cytometry.** Cells were washed with L-15 medium containing 2 mM glucose and 1% BSA, before staining on ice with Alexa647-labelled TS1/18 monoclonal antibodies. Data were acquired on a FacsCanto II (BD) and analysis was performed using FlowJo (Tree Star).

**Statistics.** No assumptions of normal distribution were made and therefore only non-parametric statistical tests were used. GraphPad Prism 6 or MATLAB R2014b was used for all statistical calculations. For multiple comparisons, Kruskal–Wallis tests with Dunn's multiple comparison correction was used. For pairwise comparisons, Mann–Whitney tests were employed. No statistical method was used to predetermine sample size.

**Data availability.** MATLAB functions for cell segmentation and leading edge detection are available on request.

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

## Acknowledgements

This work was supported by NIH Grant CA031798 (TAS). P.N. was funded by fellowships from the Swedish Research Council (VR 524-2011-891), Swedish Society for Medical Research, the Image and Data Analysis Core at Harvard Medical School, scholarships from the Swedish Society of Medicine (SLS-173751; SLS-323411) and the Blanceflor Foundation, as well as funding from the Crafoord Foundation and Knut & Alice Wallenberg Foundation. We thank Dr Brenton Hoffman for providing raw data on force—FRET efficiency relationship. We thank Dr Peter Horwath for providing the source code for CellTracker. We acknowledge the Nikon Imaging Center at Harvard Medical School for excellent help and advice on microscope instrumentation and NIH 1S10RR026549-01 for shared instrumentation funding.

## Author contributions

Conceptualization, P.N. and T.A.S.; methodology, P.N., H.E. and T.A.S; software, P.N. and H.E.; investigation, P.N.; Writing—original draft, P.N. and T.A.S.; writing—review and editing, P.N., H.E. and T.A.S.; funding acquisition, P.N. and T.A.S.; supervision, H.E. and T.A.S.

## Additional information

**Competing financial interests:** The authors declare no competing financial interests.

