## [Peer Review File · Nature Communications]

Reviewer #1 (Remarks to the Author)

In the manuscript by Nordenfelt et al. entitled "Coordinated integrin activation by actin-dependent force during T cell migration" the author describe the generation and application of $\beta 2$ integrin force sensors. In response to previous suggestions, the authors have included additional experiments, re-analyzed some data and partly rephrased the text. As a result, the manuscript has been improved and I sympathize with the authors who have performed many convincing though technically challenging experiments. The paper is interesting and original but the problem is that the lack of direct evidence to support the main conclusion of the paper ('mechanical forces activate integrins').

Major issues:

1) The main conclusion of the manuscript is that integrin activation depends on mechanical force. While the provided data are consistent with this interpretation, I do not see direct evidence for this finding. Would the experimental data not also be consistent with an alternative hypothesis in which integrins are activated by talin/kindlin binding that then allows ligand interaction leading to integrin tension? In my view, the experiments in Fig.5 and Fig.7 are unsuitable to provide formal evidence for any of the two hypotheses. Since the integrin-force statement is so central to this paper, I feel that it should be more rigorously tested. For instance, can the authors re-express mutants in talin- or kindlin-deficient cells that bind $\beta 2$ -TS integrins but do not engage the f-actin cytoskeleton and thus cannot transmit force? Such an experiment may allow the authors to disentangle effects of talin/kindlin-binding on integrin activation and force transmission. I understand that formally proving the role of mechanical force for integrin activation may be difficult, but I do not think that the current experiments are particularly helpful to settle this important (and for this paper central) issue.

2) The provided data suggest that the function of $\beta 2$ integrin is largely restored by $\beta 2$ -TS3 or $\beta 2$ -TS5 expression. Suppl. Fig. 5d shows that the re-expression of $\beta 2$ -TS3 rescues the cell adhesion defect of LAD cells, however, the data in Suppl.Fig.5e demonstrate that the migration phenotype is not entirely rescued (it seems to be restored to about 60% of $\beta 2$ -wt levels). Does this not indicate compromised $\beta 2$ integrin function? Also, why do the authors not show the same experiment for $\beta 2$ -TS5?

Minor issues:

1) The statistical evaluation of data Fig.2f has been changed and data are now compared to the C-terminal control, which I do not really understand. Is it not the difference between FN and ICAM-1 that is interesting here (and thus what should have been statistically evaluated)?

2) The suggestion that integrins can be activated by 1.5 pN (line 275) is simply not substantiated by the data. As discussed before, the authors perform bulk measurements and therefore cannot draw this conclusion. I find this statement to be misleading and do not see why the authors wish to stick with it.

Reviewer #2 (Remarks to the Author)

The adaptation of force sensing FRET methods to directly assess force on integrins reported here is likely to be of general interest. The manuscript has been edited in response to prior reviews and some new data added. Prior concerns not addressed experimentally are dealt with by addition of text to the results or discussion. This is a large body of work and while some additional questions remain I believe it is suitable for Nature Communications.

Minor correction:

Fig 5 still uses the TS8 nomenclature that has been replaced with cTerm through the rest of the manuscript.

Reviewers' comments:

Reviewer #1 (Remarks to the Author):

In the manuscript by Nordenfelt et al. entitled "Coordinated integrin activation by actin-dependent force during T cell migration" the author describe the generation and application of $\beta 2$ integrin force sensors. In response to previous suggestions, the authors have included additional experiments, re-analyzed some data and partly rephrased the text. As a result, the manuscript has been improved and I sympathize with the authors who have performed many convincing though technically challenging experiments. The paper is interesting and original but the problem is that the lack of direct evidence to support the main conclusion of the paper ('mechanical forces activate integrins').

Major issues:

1) The main conclusion of the manuscript is that integrin activation depends on mechanical force. While the provided data are consistent with this interpretation, I do not see direct evidence for this finding. Would the experimental data not also be consistent with an alternative hypothesis in which integrins are activated by talin/kindlin binding that then allows ligand interaction leading to integrin tension? In my view, the experiments in Fig.5 and Fig.7 are unsuitable to provide formal evidence for any of the two hypotheses. Since the integrin-force statement is so central to this paper, I feel that it should be more rigorously tested. For instance, can the authors re-express mutants in talin- or kindlin-deficient cells that bind $\beta 2$ -TS integrins but do not engage the f-actin cytoskeleton and thus cannot transmit force? Such an experiment may allow the authors to disentangle effects of talin/kindlin-binding on integrin activation and force transmission. I understand that formally proving the role of mechanical force for integrin activation may be difficult, but I do not think that the current experiments are particularly helpful to settle this important (and for this paper central) issue.

We don't think that the suggested experiments will work well enough with migrating cells (as talin and kindlin is required for integrin-mediated migration in leukocytes, see Renkawitz et al NCB 2009, and Svensson et al Nat Med 2009) to provide the evidence the reviewer is seeking. We agree that we cannot completely rule out alternative explanations for integrin activation and have instead modified our claims to illustrate this. See text changes in Abstract, line 20 and Main text, lines 67, 148, 223, 250, and 281.

2) The provided data suggest that the function of $\beta 2$ integrin is largely restored by $\beta 2$ -TS3 or $\beta 2$ -TS5 expression. Suppl. Fig. 5d shows that the re-expression of $\beta 2$ -TS3 rescues the cell adhesion defect of LAD cells, however, the data in Suppl.Fig.5e demonstrate that the migration phenotype is not entirely rescued (it seems to be restored to about 60% of $\beta 2$ -wt levels). Does this not indicate compromised $\beta 2$ integrin function? Also, why do the authors not show the same experiment for $\beta 2$ -TS5?

We do not claim full restoration of function (see lines 123-126). As we discussed in the previous rebuttal, we actually expected that having all $\beta 2$ integrins bear a sensor to almost completely disrupt the function. Instead what we found was that not only were these sensors expressing well and responding to force, they could even rescue cell migration to a large extent (60% of wt vs <1% for the original LAD cells), which would be the ultimate test of their function. Still, as we have argued, we believe it to be better to use the sensors as part of a larger wt protein ensemble to increase the likelihood of reporting normal function. The fact that we didn't do these experiments for all sensors is mostly time related due to the difficulties in working with the patient LAD cells and establishing variants of these. We simply picked one sensor as a proof-of-principle test of function. As we have also discussed previously, due to the very close results in all other experiments, we expect the outcome of a $\beta 2$ rescue experiment to be very similar between TS3 and TS5.

Minor issues:

1) The statistical evaluation of data Fig.2f has been changed and data are now compared to the C-terminal control, which I do not really understand. Is it not the difference between FN and ICAM-1 that is interesting here (and thus what should have been statistically evaluated)?

The leading sentence to this analysis did wrongly suggest that the relevant analysis is between FN and ICAM-1, so we understand the criticism. However, the experiment was designed as a test for differences between the constructs (compared to cTerm construct) and not between substrates, with FN being used as a negative control. The effect of substrate is tested more rigorously in other parts of the manuscript. We have now removed the leading sentence and made the section clearer. See lines 140-142, 146.

2) The suggestion that integrins can be activated by 1.5 pN (line 275) is simply not substantiated by the data. As discussed before, the authors perform bulk measurements and therefore cannot draw this conclusion. I find this statement to be misleading and do not see why the authors wish to stick with it.

We have kept the result reference (1.5 pN), but removed the speculation about activation that ended the sentence. See lines 272-273.

Reviewer #2 (Remarks to the Author):

The adaptation of force sensing FRET methods to directly assess force on integrins reported here is likely to be of general interest. The manuscript has been edited in response to prior reviews and some new data added. Prior concerns not addressed experimentally are dealt with by addition of text to the results or discussion. This is a large body of work and while some additional questions remain I believe it is suitable for Nature Communications.

Minor correction:

Fig 5 still uses the TS8 nomenclature that has been replaced with cTerm through the rest of the manuscript.

We have corrected this. See Fig. 5.